# Understanding the importance of non-material factors in retaining community health workers in low-income settings: a qualitative case-study in Ethiopia

Nikita Arora [1], Kara Hanson [1], Neil Spicer,[1] Abiy Seifu Estifanos,[2] Dorka Woldesenbet Keraga,[2] Alemtsehay Tewele Welearegay,[3] Freweini Gebrearegay Tela,[3] Yemisrach Ahmed Hussen,[2] Yordanos Semu Mandefro,[2] Matthew Quaife [1]

¹Faculty of Public Health and Policy, London School of Hygiene and Tropical Medicine, London, UK
²School of Public Health, Addis Ababa University, Addis Ababa, Ethiopia
³School of Public Health, Mekelle University, Mekelle, Ethiopia

**Correspondence to**
Nikita Arora;
nikita.arora@lshtm.ac.uk

## ABSTRACT

**Objectives** The motivation and retention of community health workers (CHWs) is a challenge and inadequately addressed in research and policy. We sought to identify factors influencing the retention of CHWs in Ethiopia and ways to avert their exit.

**Design** A qualitative study was undertaken using in-depth interviews with the study participants. Interviews were audio-recorded, and then simultaneously translated into English and transcribed for analysis. Data were analysed in NVivo 12 using an iterative inductive-deductive approach.

**Setting** The study was conducted in two districts each in the Tigray and Southern Nations, Nationalities and People's Republic (SNNPR) regions in Ethiopia. Respondents were located in a mix of rural and urban settings.

**Participants** Leavers of health extension worker (HEW) positions (n=20), active HEWs (n=16) and key informants (n=11) in the form of policymakers were interviewed.

**Results** We identified several extrinsic and intrinsic motivational factors affecting the retention and labour market choices of HEWs. While financial incentives in the form of salaries and material incentives in the form of improvements to health facility infrastructure, provision of childcare were reported to be important, non-material factors like HEWs' self-image, acceptance and validation by the community and their supervisors were found to be critical. A reduction or loss of these non-material factors proved to be the catalyst for many HEWs to leave their jobs.

**Conclusion** Our study contributes new empirical evidence to the global debate on factors influencing the motivation and retention of CHWs, by being the first to include job leavers in the analysis. Our findings suggest that policy interventions that appeal to the social needs of CHWs can prove to be more acceptable and potentially cost-effective in improving their retention in the long run. This is important for government policymakers in resource constrained settings like Ethiopia that rely heavily on lay workers for primary healthcare delivery.

## INTRODUCTION

With 24% of the global burden of disease and only 3% of the global health workforce,

## Strengths and limitations of this study

► To the best of our knowledge, this is the first study to report the perspective of leavers of community health worker (CHW) positions, to understand the drivers of their decisions.
► We provide an understanding of non-material factors influencing the retention of CHWs, which is important for policymakers to manage attrition among these workers in a cost-effective manner especially in resource constrained settings.
► We employed an iterative inductive-deductive style of analyses to allow for relevant themes to be selected, while also allowing unexpected themes to be reflected in participant narratives.
► Participants were recruited from within the country so a limitation of the study was our inability to capture the perspectives of leavers who had migrated out of Ethiopia.

countries in sub-Saharan Africa are struggling to attain universal health coverage and meet the Sustainable Development Goals by 2030.[1] In this context, over the last two decades, a large body of evidence has emerged on the importance of community health workers (CHWs) in overcoming workforce shortages and improving population health, particularly in previously underserved communities.[2–6] Although the model and scope of CHW programmes vary, these health workers are mostly female, trained for a short period on the interventions they will deliver, and usually reside in communities where they work.[7] The significance of CHWs has also been recognised in two recent reports: a WHO guideline on optimising CHW programmes[8] and the CHW Assessment and Improvement Matrix[9]—both of which recommend strategies to optimise the functioning

of CHW programmes in health systems, especially in low- and middle-income countries (LMICs).[10]

Although now seen as critical to a well-functioning health system, poor motivation and increased attrition among CHWs remains a challenge. While there is some limited evidence of effective interventions to address poor motivation and retention,[11–14] existing research has typically only explored how different, material incentive packages could improve performance.[15–17] Importantly, studies have not explored the role that community culture and group identification play in influencing CHWs' preferences, motivation and retention in the long term. In particular, no study to date has studied CHWs who have left the health system to understand why they left, and what could have averted their exit.

In 2003, Ethiopia launched the health extension program (HEP), a primary healthcare delivery strategy designed to make up for the low number of doctors, nurses and midwives. HEP has focussed on delivering essential healthcare services using lay CHWs called health extension workers (HEWs), mainly targeting agrarian communities.[18] HEWs complete a year-long training in delivering primary healthcare interventions like family planning services, latrine construction and basic preventive and curative services for communicable and non-communicable diseases.[19] Unlike CHWs in many other countries, HEWs are salaried government employees with currently more than 40 000 workers deployed in the country.[18 20 21] HEP has recently been recognised by WHO as a role model for global CHW programmes, due to its focus on integrating CHWs in the health system as civil servants; training HEWs for a significant period of a year before deployment; and offering educational opportunities to upgrade to higher levels of the health workforce.[8]

A recent, national evaluation of HEP had found the overall job satisfaction of HEWs to be quite low. More than half of the study sample reported to be unsatisfied with their current posts, suggesting that their retention could be affected in the long run. These apprehensions were substantiated by data indicating a gradual rise in the rate of attrition among HEWs over the programmatic lifetime of HEP, between 2005 and 2019. The average annual rate of attrition was reported to be close to 3%, with overall attrition since the start of the programme being 21%.[22] This showed a clear rise in HEW attrition since the last national assessment of HEP published in 2011, which estimated overall attrition in the cadre for the period between 2005 and 2010 to be 6.5%.[23]

HEWs take up a large proportion of the Ethiopian health budget; 21% of the recurrent health expenditure in 2010/2011 was spent on HEW salaries,[21] and so it is critical to make sure that experienced HEWs are retained over time to use this budget efficiently but also to sustain the delivery of quality healthcare. Yet, to date few studies have researched why HEWs leave their posts. Most research has sought to identify financial and non-financial incentives, which motivate HEWs.[24–29] Some ethnographic accounts of HEWs have also studied

the context in which they work,[30–32] and more broadly, research has been conducted on contextual factors influencing the performance of CHWs.[26]

While material incentives that align with the preferences of CHWs are relevant to studying retention, behavioural theories like the social identity approach have seldom been applied to empirical findings, to account for the social behaviour of health workers. This approach studies the social identity, context in which they work, along with self-efficacy and outcome expectancies that could influence their labour choices to stay in or leave their jobs. Further description of this approach is provided in the discussion section. Moreover, previous studies have never researched the perspective of CHWs who have left these positions ('leavers'), to capture the drivers of their decisions.

This study identifies factors influencing the labour market decision of CHWs in Ethiopia to leave or stay in their jobs, taking the perspectives of current HEWs, leavers and the health system. Furthermore, we use the data generated from qualitative interviews to demonstrate how group identification can also influence the social behaviour and preferences of HEWs towards working conditions that ultimately influence retention in the health workforce. This evidence makes an original contribution to the global literature on retention of CHWs, as countries gear towards strengthening their own CHW programmes.

## METHODS

We conducted this study between January and August 2019. In the first stage, we undertook a literature review to identify conceptual frameworks that link motivational factors to health worker retention. We adapted the conceptual framework by Ormel *et al* 2019,[33] shown in figure 1 which critically analyses the use of a mix of incentives and their relationship with CHW motivation and work behaviour, to include the role that pro-social preferences play towards prioritising non-material incentives. This model was most in line with our study objectives and thus selected to inform our interview topic guides and

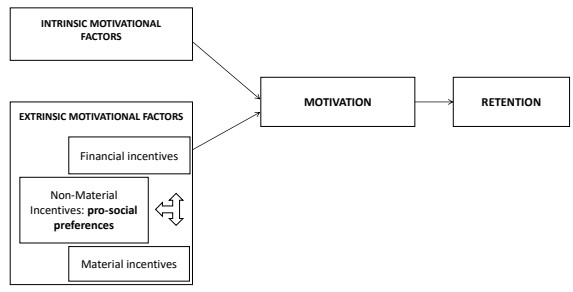

**Figure 1** A framework of relationships between motivational factors, motivation and community health worker work behaviour. Modified from Ormel *et al*.

**Table 1** Interviews conducted per informant type

| Study population | Active HEWs | Leavers of HEW positions | Key informants |
|---|---|---|---|
| Sample size | 16 (8 per region) | 20 (10 per region) | 11 (5 per region, plus one from Federal Ministry of Health) |
| Gender | All females | All females | 3 females, 8 males |
| Marital status | 10 married, 4 single, 2 divorced | 8 married, 1 divorced, 1 single | 10 married, 1 single |
| Purpose of inquiry | To capture their perspective on factors affecting HEW motivation and labour choices | To understand factors influencing decisions to leave | To capture the perspective of key stakeholders and identify policy levers that could be modified to improve HEW retention |
| Sampling technique | Maximum variation sampling from a list of HEWs working in study districts, for diversity of age, geographical location and years of experience | Snowball sampling | Purposive sampling, with variation in administrative levels, seniority and level of engagement with HEWs |

HEW, health extension worker.

categorise our qualitative data described in the section below. Following this, we conducted qualitative narrative research with HEWs, leavers and policymakers.

### Study setting

Qualitative indepth interview data were collected from four districts in two Ethiopian regions, between May and July 2019. Raya Azebo and Saharti Samra districts were covered in Tigray, and Dilla Zuria and Silte in Southern Nations Nationalities and People's Republic (SNNPR). HEWs were located in a mix of urban and rural backgrounds. The regions were purposively sampled to capture the varying extent of HEP implementation and thus HEW retention in different regions, which was likely to differ due to the political set up in Ethiopia. Historically, Tigray has been a better performing region on health indicators, in comparison to SNNPR, and so we expected variation in perspectives from staff working in the two regions.

### Sampling and participants

Data were collected from three key populations: active HEWs, leavers and key informants (KIs). Key informants were policymakers at the national, regional and district level. Details on sampling and respondents per region are presented in table 1.

We interviewed respondents until sufficient saturation was attained in the themes emerging from interview data. A total of 47 semi-structured interviews were conducted. The mean age of respondents across all three study groups was 31 years, ranging from 24 to 40 for leavers, 20 to 49 for HEWs and 26 to 48 for KIs. The mean time worked in the health system was 6 years for HEWs, ranging between 1 and 13 years and 7 years for key informants, ranging from 0.5 to 8 years. Leavers had spent on average 6 years in their current jobs, ranging from 1 to 9 years. Key informants included HEW supervisors, senior officials at district health offices, HEW experts at district levels, HEW coordinators at regional levels and a senior official at the HEP directorate, Federal Ministry of Health.

### Conduct of the interviews

Separate topic guides were drafted for each study population, informed by literature on factors affecting the motivation and labour choices of CHWs in LMICs, and the framework by Ormel et al.[33] Key topics covered in the interviews included reasons for choosing the HEW profession, motivating factors, challenges faced in their jobs and preferences towards job attributes. For leavers, we additionally inquired about their reasons for leaving. Topic guides were piloted and pre-tested with interviewers and members of the study populations. Research assistants experienced in qualitative research conducted interviews after receiving 2 days' training on study aims, topic guides, ethics of research and its conduct. Training included how to identify and reduce social desirability tendencies in respondents. All research assistants were Ethiopian women between ages 24 and 35 years. Respondents above the age of 18 and willing to participate were approached by research assistants through telephone calls. Principles of confidentiality and informed written consent were upheld during interview administration, in compliance with the ethical approval conditions of the project. Each interview was conducted in the language local to that region, in private spaces—normally at the back side of the health post where the respondent and interviewer could be left alone. As much as possible, the research assistants held interviews when HEWs were comparatively less busy with work, and took on average forty-five minutes to complete. All interviews except two were audio-recorded, translated and transcribed in English by interviewers, who also took notes and discussed in daily debriefing sessions between researchers. For the two interviews where audio recording was not possible due to respondent refusal, research assistants took down

detailed notes which were later used for analysis. Each interview was assigned a unique alphanumeric identification code during analysis, and personally identifiable information was removed from transcripts during coding.

### Data analysis
Transcripts from the audio recordings were analysed using an iterative, inductive-deductive approach[34] in NVivo 12.[35] Themes were identified by 1: reading and re-reading the transcripts and making notes on relevant issues 2 ; listing out these issues in the form of a codebook and attaching codes to relevant sections; and 3 writing narrative summaries of relevant themes and subthemes that emerged most frequently and/or were appropriate to the study.[36]

### Patient and public involvement
This study focussed on capturing the perspectives of CHWs and policymakers, and was undertaken without the involvement of patients.

### RESULTS
Since leavers were the key population of interest in the study, we first describe the jobs that they were engaged in at the time of interview and summarise factors that sped up their exit from HEW positions. Key factors influencing HEW motivation and labour choices are then presented in detail.

### Leavers' destinations
We found significant variation in the type of employment that the leavers were currently engaged in. Migration to non-health jobs in the Middle East due to better pay was a particularly unanticipated finding. Notably, no participant reported leaving or wanting to leave Ethiopia to work in a health system abroad, perhaps because HEWs are relatively low skilled by international standards.

> Most HEWs think about going to foreign countries. Like Arab countries. The salary in Arab country is relatively good in comparison with HEWs. […] They think, here the workload is very high and the salary is very low so, why don't I go to an Arab country? And why don't I change my life in a short time? - Active HEW, SNNPR

A significant number of leavers reported to have become full-time homemakers. 'I raise my children, I am a housewife' said a leaver from Tigray. Some women were currently self-employed and owned small businesses highlighting increased earnings and autonomy. 'I have my own grocery. If you are an excellent worker, it provides higher income. It also has freedom; nobody can come and shout at you' reported a leaver from Tigray. Other respondents remained working in different health system roles, including as lab technicians and administrators in government health facilities.

### Catalysts influencing exit: pro-social preferences
While HEWs did not generally anticipate leaving their jobs, they did leave when they felt they had lost the appreciation of their community or supervisors. These were the two main reasons for leavers to finally quit their jobs, despite other challenging working conditions reported by active HEWs and leavers alike. We call these factors *catalysts,* or the triggers that sped up the process of attrition.

#### Failure to receive support and validation from supervisors and senior staff
Conflict with supervisors and senior managers was the main reason why leavers claim to have quit. These 'conflicts' often seem to have started with a senior official disrespecting the HEW, resulting in a negative shift in their status, social standing and esteem and thus in their identity as a HEW. Supportive supervision, with appropriate acknowledgement and validation from their managers was identified as a critical factor in the retention of HEWs in Ethiopia.

> …the director came to my home and insulted me when I was very sick. He said this institution is neither your mother's nor father's; either perform your job appropriately or leave. I immediately left my job, and didn't even take my monthly salary - Leaver, Tigray

#### Reduced acceptance and validation from the community
Another key element for retention was receiving respect, acceptance and validation from the community for whom the HEWs worked. Despite tough working conditions, the opportunity to improve community health attracted many to their jobs. A negative shift in their social identity, due to low community acceptance, influenced working conditions and status and their exit.

> There is no appreciation from the people in my woreda (district)… always they will criticise the HEW and service delivery… they are fault finders. - Leaver, SNNPR

### Key factors influencing HEW motivation and retention
Numerous factors reported by HEWs, leavers, and KIs were identified as those influencing HEW motivation, and retention in the workforce. Using our conceptual model from figure 1, we classified these into two categories: Extrinsic and Intrinsic.

### Extrinsic motivational factors
#### Financial incentives
Financial incentives in the form of salaries or wages were found to be important among active HEWs as well as leavers. Current salaries were not considered to be commensurate with workload, their compensation not being enough to cover monthly household expenditure.

> HEWs do many overlapping tasks, but salary doesn't balance the work we do…the salary does not reflect living conditions of HEWs. Since we don't have

additional income, and spend all our time at work, it's difficult to live on our existing salary. - Active HEW, SNNPR

Key informants, including HEW supervisors and senior officials at health centres unanimously agreed that HEW salaries were inadequate. HEWs cater to a large population, often in topographically difficult terrains and on foot, so physical strain due to their job came up as a common theme and a constraint to their motivation. 'I still remember how horrible it was…the 4-hour walk in the mountains. It rains over us, and the sun burns so bad', remembers a leaver from SNNPR.

## Material factors

Material factors were also seen as being important in influencing HEW motivation and retention. These were often driven by whether adequate drugs, equipment and infrastructure were available at the health post. Such factors were found to be critical not only to support their daily work, but important to sustain the rapport and confidence the community had in them by managing to do the tasks entrusted to them. Sometimes facilities were perceived so lacking that faith was the only answer:

Sometimes I support the labouring mothers by praying to Gebriel (Angel), because what we learn is different from what we apply. The materials that we have are inadequate; we only have delivery kit, which contains scissor, and cord tie. When a mother delivers at hospital, many things are provided to her and her baby, but here we have nothing to give her. - Leaver, Tigray

In addition, HEWs and leavers suggested that material incentives such as motorcycles for transportation should be provided as part of their work package, to decrease their physical burden.

Furthermore, the gender of HEWs results in a double burden, as many mothers with infants mentioned that it was hard for them to do their daily tasks as a HEW, alongside caring for their infants.

It is very difficult having a child. I leave from my home early morning at 6 am […] I may stay up to 6 pm, sometimes I don't even have time to drink water after coming back from field work. So, imagine doing all things having a baby - Leaver, Tigray

For HEWs with young children, the absence of childcare was a disincentive to continue in their jobs after giving birth.

HEWs also mentioned not always feeling safe in travelling to rural areas. 'Facilities like motor for transportation should be fulfilled. This security issue also needs attention since in rural areas females can be abused,' stated a leaver from Tigray.

## Non-material factors

Most importantly, HEWs and leavers mentioned highly valuing the non-material factors such as appreciation from their communities and supervisors. The opportunity to improve community health, especially that of mothers and children, and gain their community's trust, respect and acceptance, was unanimously described as the top factor motivating them to stay in their jobs.

When I get the acceptance of healthy mothers and children, I am satisfied. Otherwise, the salary is not enough; the high workload is as I told you before. - Active HEW, SNNPR

Sometimes HEWs were not as easily accepted by their community, which demotivated them. Often respondents claimed that these demand-side barriers existed because of low levels of education and awareness among community members, which also led them to reject healthcare interventions such as family planning and latrine construction.

The community's behaviour is difficult. For example, when we go to their home to educate them about environmental hygiene, they may close their door and leave from home. They say, oh! She is coming! When I enter through the front door, they will leave the house from the back door. It is for them but they do not understand. To teach them about something we will take many days. They have a shortage of knowledge. - Leaver, Tigray

Other non-material demotivating factors were things that HEWs and leavers identified as lacking in HEW jobs. For example, the placement of HEWs in health posts, often far away from their hometowns where their husband and children are based, limited their motivation and retention. All three study populations agreed that the absence of opportunities to transfer to a facility closer to their family was frustrating, unfair and led HEWs to leave their positions. 'This was my main reason to leave my job… Imagine that you can't meet with your husband as well as your children for a long time because there is no transfer (opportunity)', mentioned a leaver from Tigray.

In addition, respondents reported concerns around the ways in which they could progress in their careers. The majority of HEWs are currently hired as level 3 (nowincreasingly level 4) health workers and according toHEP, HEWs have the opportunity to upskill to the nextlevel after taking a competitive exam. After this, based onopportunities available and skills needed in the district,HEWs can further upgrade to diploma level courses insubjects like midwifery, and even complete a master'sprogramme in public health from government universities. Two key issues around career progression were identified. First, HEWs that were keen to upskill to the next level had to take this competitive exam in English—a language they are not generally proficient in and do not normally use in their jobs, and on topics in which they had not received enough training. The success rate for these exams was thus found to be low. HEWs complained that while many of them are excellent field workers with many years of experience in delivering healthcare, their inability to

do well in an exam should not be the sole determinant of career progression.

The second key issue was for HEWs who did manage to upgrade to the next level, but despite upskilling, were expected to return to their old jobs at the health post. Many HEWs agreed that while after upgrading, their remuneration increased (or was expected to increase in the following months), they were expected to undertake the same tasks in the same health post as before.

> After we get back from our level 4 study, we will be placed to the same kebele (village) as before. We need to be refreshed, be in a new place! Alongside with transfer, we should also be assigned to health centres (promoted to a higher health facility). - Active HEW, SNNPR

Another reason why HEWs and leavers felt de-motivated was the lack of support, oversight and acknowledgement from supervisors and managers, who said that supervision was based on a model of faultfinding, not mentorship.

> …[…]. I was so tired that night that I could not clean all the blood and every mess (after single-headedly doing a delivery at the health post). Next morning the woreda (district) officials showed up and insulted me without considering what I have been through. It was so painful not to be understood to this level. - Leaver, SNNPR

### Intrinsic motivational factors

Many HEWs mentioned that the key reason for joining the profession was to serve the community where they were raised.

> Most of the time in our environment, the mothers don't use contraceptives, they don't give birth in health centres and they don't get antenatal care. The mothers normally give birth in their home with a traditional birth attendant. Because of this, many mothers die. When I saw these types of problems in my community, I decided to become a HEW. - Active HEW, SNNPR

Some HEWs also insisted that financial incentives were less important than intrinsic factors and that the profession requires women to be truly dedicated to the community's health improvement, to survive in their jobs.

Many leavers mentioned having left their jobs out of frustration with challenging conditions but confessed to have really enjoyed working towards improving community health. 'Regarding the profession, health extension work itself has no concerns. I believe as a HEW you get to serve or work for the community which is great… it's the working conditions that are problematic,'said a leaver from Tigray.

### DISCUSSION

Our study findings from two regions in Ethiopia contribute new empirical evidence to the global knowledge and debate on factors influencing the motivation and retention of CHWs in LMICs. It is the first study to include the perspectives of those who had left their posts. Since we wanted to capture the individual experiences and behaviour of all three respondent groups, we refrained from conducting focus group discussions and committed to using in-depth interviews. Moreover, we believe that reasons why people leave their jobs or things they find unsatisfying are of a sensitive nature, unlikely to be disclosed in a group of peers.

Many of the extrinsic motivational factors we identified, such as wages and allowances, were similar to those identified for CHWs in other settings.[11 24 27 33 37 38] For example, a study in Bangladesh reported lack of time to attend to their own children and other household responsibilities, insufficient profit/salary and their families' disapproval as reasons cited by CHWs for leaving their posts.[39] In Nigeria, village health workers reported low work satisfaction due to the lack of career advancement opportunities, low salaries and poor supervision.[40]

Our study offers a number of new perspectives that we believe are valuable in Ethiopia and in other LMICs. Discussion around an adequate career path for CHWs in LMICs is ongoing.[8 33 37] Despite WHO's repeated recommendations on a set career ladder for CHWs to be established in individual country contexts,[8] the uptake has been low by governments. For example, in Ethiopia, there is evidence that the majority of HEWs are keen to take on more responsibility and upgrade to become nurses, pharmacy technicians and health administrators[41] but no such career path is offered to them. In another study, access to and provision of upgrading and promotion opportunities was identified as one of top five measures that can motivate HEWs and improve HEP services.[42] The need for, and the value of, career progression among CHWs to improve job satisfaction was also a key topic of discussion at an international symposium on CHWs in 2019.[43] Moreover, the lack of educational opportunities and poor career development seems to be a bigger cause of concern in Ethiopia, as similar factors also drove up the rate of attrition for higher level professionals like doctors and nurses by nearly three times, in comparison to other allied health professionals lower in the hierarchy[23].

Other material and non-material incentives affecting retention in this context were better living and working conditions that included their ability to live close to their family and have easy access to water, electricity at home and at work. According to a study published in 2007,[28] the living and working conditions of HEWs during early stages of HEP had not met basic standards. A more recent study suggested that many health posts were still missing basic infrastructure like water supply, electricity in 2012.[44] The mean availability of tracer items for basic facilities, infection prevention, malaria diagnosis and essential medicines at health posts was 37%, 29%, 52% and 47%, respectively, according to data from a service availability and readiness assessment, in 2016.[45 46]

Additionally, there is growing recognition of the importance of gender inclusiveness and equity in healthcare,

which entails transforming the systems within which women work, such as highlighted in a recent report from the WHO's Gender Equity Hub.[47] In a Cochrane review, socio-cultural norms that restrict movement of female CHWs and govern acceptable male-female communications were also identified as barriers to doing their jobs successfully.[48] Jackson *et al*[49] apply a gender lens to HEP and to the role of HEWs and conclude that by changing gender norms and reducing constraints to gender equality, HEP could have more transformative outcomes not just for HEWs but for the communities they serve. Since the majority of HEWs in Ethiopia are women of reproductive age, providing them with childcare, particularly for when they are away for house visits, could be a step forward in gender transforming their work environment.

## Social behaviour and preferences of HEWs

While our evidence supports the importance of material incentives, we also identified other influences on social preferences of CHWs, which could help understand how they prioritise across multiple factors. Such insights could inform the development of new interventions to motivate and satisfy CHWs and retain them in the long term.

In this context, while conventional models have identified motivation as intrinsic and extrinsic, our empirical results identified two further additions—pro-social preferences as a non-material motivator, and social identity as a factor that could influence how CHWs trade among attributes. The social identity approach demonstrates how processes within an individual that influence behaviour are dependent on interpersonal relationships and group memberships, as well as their perceived value and significance to the individual.[50 51] This approach states that when a person identifies as a member of a group, and when a given group identity is relevant to an individual, their behaviour becomes more focussed towards what is seen to be in the group's interest, rather than their own.[51 52]

Thus, when workers define themselves in terms of a personal identity it could be expected that individual motivators such as personal advancement and financial incentives may be more influential. However, when defining themselves in terms of a social identity, motivators that impact on the group one identifies with, such as their status, standing and acceptance in the group may become more influential,[53] like in the case of HEWs. This is a hypothesis that merits further empirical investigation. While the social identity approach is increasingly being applied in high-income countries,[11 53] it is less common in LMICs. To our knowledge, the inSCALE project, which operated in Uganda and Mozambique,[11] is the first to use the social identity approach in a LMIC context to address these constraints in motivation of CHWs. Our study drew on formative research results from the inSCALE study and applied the social identity approach for establishing links between identification and motivation[54 55] in the context of CHWs in Ethiopia.

It has already been recognised that non-material interventions to support CHWs can contribute substantially in creating a more satisfied health workforce that is able and willing to continue delivering quality healthcare to communities.[31 49 56 57] In the Ethiopian context, focus particularly could be accorded to improve not just the availability of strategic resources such as mentoring and supervision, but the quality of support offered by often male supervisors to these female workers. Addressing HEW aspirations to progress in their jobs by providing sufficient upgrading opportunities, tailored to their preferences and abilities, has good potential for improving their job satisfaction, reducing attrition. Clearly, positive community attitude towards HEWs is a key demand-side requirement for HEWs to stay motivated. We believe a good rapport between HEWs and the community often results when HEWs are capable of providing healthcare to the standards expected by the community, which is a function of having health posts equipped with adequate infrastructure as well as well-trained HEWs. Equally important is that HEWs are emotionally satisfied, not having to live apart from their families due to the lack of transfer opportunities.

Future research should explore the development of interventions that can create and maintain trust between CHWs and the community. It could further be evaluated if a bottom-up approach that is designed with the inputs of CHWs and the community, is better tailored to the needs and realities of both.[25] In addition to health outcomes, policymakers should also invest in studying outcome measures such as competencies and self-esteem of health workers as this can have direct effects on their retention and indirect effects on the sustainable delivery of population health.

## CONCLUSION

Our study showed that CHW jobs in LMICs including Ethiopia continue to be challenging, and incentives that align with their preferences have the potential to improve their motivation, influencing retention. However, modifying material incentives alone might not improve retention in the long term. Using empirical data from our study and theories of CHW motivation from the literature, we have demonstrated that CHWs identify themselves as members of a group (in this case their community and team). Thus, appealing to their social needs may represent a relatively more acceptable, potentially cost-effective and complementary strategy to the traditional approach of using financial incentive packages for improving retention, particularly in the long run in resource-constrained settings. These non-material factors are important to be considered by government policymakers in resource constrained settings like Ethiopia that are struggling with critical health workforce shortages and inadequate health budgets. The voices of health workers can offer insights that may otherwise be missed and should thus be included while designing programmes to improve retention.

**Acknowledgements** The authors would like to thank all the respondents who participated in the interviews.

**Contributors** All authors were involved in the original design of the qualitative study in Ethiopia. ATW, FGT, YAH and YSM conducted and translated all the interviews. DWK and ASE provided extensive in-country support. NA was the principal investigator who oversaw the fieldwork and conducted majority of the analysis, reviewed and approved by MQ and KH. NS provided expert guidance on manuscript development and analysis. All authors read and approved the final manuscript.

**Funding** This study was funded by the Wellcome Trust (grant 212771/Z/18/Z). Data collection was partly funded by IDEAS—Informed Decisions for Actions to improve maternal and newborn health (http://ideas.lshtm.ac.uk), which is funded through a grant from the Bill & Melinda Gates Foundation (BMGF) to the London School of Hygiene & Tropical Medicine. (Gates Global Health Grant Number: OPP1149259). The funder had no role in the study's design or conduct, data collection, analysis or interpretation of results, writing of the paper, or decision to submit for publication.

**Competing interests** None declared.

**Patient and public involvement** Patients and/or the public were not involved in the design, or conduct, or reporting, or dissemination plans of this research.

**Patient consent for publication** Not required.

**Ethics approval** Ethical approval was obtained from the London School of Hygiene & Tropical Medicine, UK (ref. no. 16177), as well as Addis Ababa University, Ethiopia (ref. no. 015/19/SPH) in March 2019.

**Provenance and peer review** Not commissioned; externally peer reviewed.

**Data availability statement** Data are available upon reasonable request. The data sets generated and analysed in the study are available on reasonable request made to the corresponding author.

**ORCID iDs**
Nikita Arora http://orcid.org/0000-0001-5123-7751
Kara Hanson http://orcid.org/0000-0002-9928-2823
Matthew Quaife http://orcid.org/0000-0001-9291-1511

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
