## [Reviewer comments · BMJ Open]

ARTICLE DETAILS

TITLE (PROVISIONAL)	Understanding the importance of non-material factors in retaining community health workers in low-income settings: A qualitative case-study in Ethiopia
AUTHORS	Arora, Nikita; Hanson, Kara; Spicer, Neil; Estifanos, Abiy; Keraga, Dorka; Welearegay, Alemtsehay; Tela, Freweini; Hussen, Yemisrach; Mandefro, Yordanos; Quaife, Matthew

VERSION 1 – REVIEW

REVIEWER	Ruth Jackson Honorary Fellow School of Humanities and Social Sciences Faculty of Arts and Education Deakin University Australia
REVIEW RETURNED	24-Mar-2020

GENERAL COMMENTS	Understanding the importance of non-material factors in retaining community health workers in low-income settings: A qualitative case-study of Ethiopia This study on retention rates of community health workers in low-income settings is through a qualitative case study in two regions in Ethiopia. As the role of qualitative research is to help us better understand human behaviour and its context – in this case, the research question seems to be what motivates female Health Extension Workers' retention and labour market choices. Should they stay or go? And if they choose to go, where can they go, given the limited work opportunities in rural kebeles in Ethiopia? HEWs' retention/attrition rate is an important issue in Ethiopia given the number of HEWs employed and the essential health services they provide at the community level.  In the title, should it read ...A qualitative case study in Ethiopia rather than of Ethiopia? In the Abstract spell out Health Extension Workers the first time you use it. Although you provide the total number of HEWs, can you provide evidence of how many new HEWs are being trained each year and what the attrition rate actually is? How many are being offered upgrading training to Level IV each year? Attrition is a significant problem but you don't quantify it. p. 7. Where in Tigray and SNNPR was the study conducted? Both are very large regions with considerable geographical differences. Even if you don't want to name the woredas, perhaps indicate the zones or give some idea where the study was located in the regions. Were HEWs from rural, remote or urban kebeles/woredas? There's a lot of research on the role and performance of HEWs, but far less research on the experiences of HEWs themselves. Are you able to provide a bit more detail and description about the HEWs? How far was the health centre from the health post where they worked? Were they supported by the Health Development Army in their
---

area or did they find it added to their workload? Has the HDA strengthened HEWs relationship with the community? What about expectations about being involved in politics? Being on the kebele council? Just a couple of extra sentences perhaps so reader get more of a feel about them.

6. On the role of gender, female HEWs are subject to the same social hierarchies that prevail in the wider society so relying on female HEWs may replicate or reinforce gender norms and biases in communities and within health systems. (Maes K, Closser S, Vorel E, Tesfaye Y. Using community health workers. *Discipline and Hierarchy in Ethiopia's Women's Development Army Ann. Anthropol Pract.* 2015;39(1):42–57) and see note 8. Consider whether you should mention the top-down nature of Ethiopia's HEP, and the fact that health workers such as HEWs are expected to remain 'subordinate'. Were HEWs free to move around?

7. You could add that you tried to avoid/reduce social desirability bias (in addition to) conducting interviews in private and by ensuring HEWs understood that interviews were confidential. It is well known that responses can often be 'in line with government policy' or that respondents may not want to raise negative or politically sensitive issues.

8. A paper you may have missed is this one:
Jackson, R., Kilsby, D., & Hailemariam, A. (2019). Gender exploitative and gender transformative aspects of employing Health Extension Workers under Ethiopia's Health Extension Program. *Tropical Medicine & International Health*, 24(3), pp. 304-319. doi:10.1111/tmi.13197

Although the focus is on gender, much of the data are similar. The paper also helps provide a better understanding of the working conditions of HEWs, lack of opportunities for training, safety, recognition, trust and respect from community members, and reasons why HEWs want to leave their positions.

9. Study limitations. Can the outcomes of the study be generalised to other regions in Ethiopia? Or other sub-Saharan countries? If not, you can mention that generalisability wasn't the intention but that your study is unique as it captures the voices and experiences of HEWs.

10. Page 3, 'Interviews were audio-recorded, and simultaneously translated into English and transcribed for analysis.' This sentence needs tweaking: Interviews were audio-recorded and then simultaneously ...

11. Page 9. Line 182. Just to be clear, They were all women...I'm sure you mean the research assistants but there's a lot of going back and forth between research assistants and respondents in this paragraph and I had to read it again to be sure so consider if you need to clarify.

You don't mention how you recorded information for the two interviews that were not audio recorded. Did you take notes?

In my experience, health posts are not necessarily private spaces as people are often coming and going to see the HEWs and there are usually two HEWs assigned to each health post. How did you ensure the health post was a private space for you to interview one HEW? How did you select one HEWs at a health post (and not the other, or did you interview two HEWs at the same health post?).

12. Can you mention the attrition rate of other health professionals? I wonder if you can compare the attrition rates compared to other health staff. Perhaps doctors have more opportunities to go overseas? The 2010 Human Resources for Health Country Profile: Ethiopia, though out of date showed that that higher levels of the health workforce (such as general practitioners and specialists) are mainly dominated by males and nurses, midwives, and HEWs are predominantly female so is it gendered?

I have heard about HEWs going to work in the Middle East before, but given the clampdown on this by the government, I wonder just how many opportunities they might have to work elsewhere, unless they move to an urban centre.

There are a couple more references you may have missed and might like to consider here or in future publications for comparison.

This one looks at deployment and attrition of health workers in East Wollega:

<https://www.ncbi.nlm.nih.gov/pmc/articles/PMC3275894/>

Although this paper is from 2007, but you could make a comparison to your results as it mentions future aspirations of HEWs:

https://www.researchgate.net/publication/242466722_Study_of_the_Working_Conditions_of_Health_Extension_Workers_in_Ethiopia

Very few expect to stay in the kebele of their present assignment or even as a health extension worker for more than two years. Only 16% expect to stay more than three years. The majority would like/expect to upgrade to nurse (about 70%) and the rest to environmental health. Few mention pharmacy technician, administrative positions and, interestingly, two mentioned upgrading to diploma in HEW. None chose to move to private/NGO health services, working outside the health sector or stopping employed work altogether.

You might want to consider these reports as well (although it appears the links are no longer working you might be able to find them):

Center for National Health Development in Ethiopia, & Columbia University. (2011).

Ethiopia Health Extension Program Evaluation Study, 2007-2010, Volume-II. Health post and HEWs performance Survey. Addis Ababa: <http://www.cnhde.org.et/wp-content/uploads/2013/04/Part-II.pdf>

From the report:

HEWs were asked to identify measures that could motivate HEWs and improve HEP services. The top five suggestions were: access to and provision of upgrading/promotion (57.5%), adequate supply of logistics (29.8%), salary increment (29.5%), access to transportation (15.4%), and regular supportive supervision (13%). Other factors suggested include refresher training and provision of housing.

And this one:

Center for National Health Development in Ethiopia, & Columbia University. (2011).

Ethiopia Health Extension Program Evaluation Study, 2005-2010, Volume-IV. Support and management of HEP. Addis Ababa, Ethiopia: <http://www.cnhde.org.et/wp-content/uploads/2013/04/Part-IV.pdf>

Out of the total 3,241 HEWs deployed in the 64 woredas since HEP implementation, which varies between one to 6 years of implementation in the sample woredas, a total of 212 HEWs left their HEP work with overall attrition rate of 6.5%. The main reasons for leaving their HEP work in the woreda were: changed field of work (71 HEWs), due to personal reasons such as marriage and illness (68 HEWs), and due to uncomfortable work environment such as remoteness of kebeles, high workload, and low remuneration (31 HEWs).

The total number of HEWs deployed since HEP was launched (one to five years of implementation period) in the 64 woredas surveyed was 3,241. The average number of HEWs deployed was 51 per woreda. Over the period of HEP implementation in the sample woredas, a total of 212 HEWs left their HEP work in the woreda with an average of 3 HEWs per woreda. The overall attrition rate was 6.5% over the program period. It should be noted that the average duration of implementation in Tigray and Dire Dawa is 5 years, and it is 4 years in Amhara and SNNP. In Oromia it is 2.7 years. Thus, the overall attrition rate of HEWs per year is about 2% (6.5% over 3.5 years).

Among the 212 HEWs who were reported to have left from assigned kebeles in their respective woreda, nearly a third (71 HEWs) left their kebele because they changed their field of work. An equivalent number (68) of HEWs left their job for personal reasons including marriage and medical reasons. The reason for some HEWs (31 HEWs) was reported to be uncomfortable work environment such as remoteness of kebele,

	workload, and low remuneration. The other reasons were because they were transferred to another woreda and dismissed due to discipline reasons. 13. Recommendations. Are you able to make recommendations to policy makers ‘that appeal to the social needs of CHWs could represent a simpler and more cost-effective means of improving their retention’ (as you’ve stated in the Abstract) or about any of these any of these factors: as Maes et al. 2019 have done: fair pay levels, opportunities for advancement, and representation in high-level policy decisions for would both motivate and truly empower them; or equal access to strategic resources such as mentoring and supervision, administrative and infrastructural support, secure funding sources and employment contracts and networking, as Jackson et. al 2019 have done? Should HEWs be from the community that they serve? What about task shifting to the Health Development Army and the impact of this on the workload of HEWs? Has that helped increase acceptance and validation from the community? How can HEWs be better supported by their supervisors (who are often male)? When getting to know, working with and interviewing HEWs over many weeks (and there were some HEWs I’d met years earlier doing other research), I sensed that once they had come to a decision to leave, they were going to leave no matter what, even if it meant for example, fulfilling the obligation to work for another year or two to fulfil their upgrading conditions. Awhile back I ran into a HEW who had told me she was ‘definitely’ going to leave, but she had finally managed (after years of requesting) to get a transfer to another kebele after she completed Level IV training. Another HEW had left to set up a shop in town with her family (she was from the original batch of HEWs from 2004, had worked for well over 10 years and was a bit older than the others), and another left to live with her husband in another woreda as the choice between being a HEW and being able to live with her husband and family became too difficult and she couldn’t get a transfer. 14. Future research. Can you make some suggestions for future research? A couple of suggestions could come from Maryse Kok’s publication, Performance of Community Health Workers: Optimizing the benefits of their unique position between communities and the health sector. She states that the study ‘did not find many mechanisms related to feelings or behaviours of actors in the health sector that led to trusting relationships between them and CHWs. This could be an interesting topic for further research, to input into the development of interventions that stimulate trust between CHWs and actors in the health sector.’ She goes on to say it would be interesting to ‘evaluate whether programmes that are designed with input from CHWs and community representatives are better tailored to the realities and needs of communities than programmes that are designed in a more top--down way.’ She also suggests that attention to ‘outcome measure, such as competencies or self--esteem (presumably much more complex) is needed, preferably combined with outcomes at the level of the end--users or impact.’
--	---

REVIEWER	Sonia Hines Centre for Remote Health, Flinders University, Australia
REVIEW RETURNED	31-Mar-2020

GENERAL COMMENTS	Thank you for this well written paper on an important problem. Line 117 The link between work satisfaction and retention has not been well made. The authors, if at all possible, should use actual figures of vacant positions, proportions of positions vacated over time, or other standard retention measures to illustrate the problem and justify the conduct of this study.
--

	Lines 200-365 Overall the results section is well written and clear, but I think it would benefit with greater focus on participants' voices as illustrations of the findings the authors are making. Additionally, it would be useful to have a slightly more detailed description of each speaker being quoted, so the reader can get a sense of the variety of perspectives being shared. Several of the factors being reported are not being explicitly linked to retention by the authors. Financial incentives, material factors, and non-material factors in particular are presented like factors influencing job satisfaction and do not link to why the participant left or is considering leaving. A revision of the results section to refocus on the purpose of the study would be helpful. Line 377 Referring to this research as 'case study' is inaccurate. Line 425 The point about the INSCALE project should be expanded to explain how it relates to this study.
--	--

REVIEWER	Indrani Saran Boston College School of Social Work, United States
REVIEW RETURNED	02-Apr-2020

GENERAL COMMENTS	Summary: This study uses qualitative methods to identify some of the factors that affect community health workers motivation in their jobs in two regions in Ethiopia. As the authors point out, CHWs are increasingly a key component of the health system in many countries and are being used to deliver basic health services to communities. Thus, a better understanding of motivation in their role is crucial to both retaining this crucial workforce as well as maintaining high quality of services. Strengths: There are several strengths of this study: - The authors did interviews both with current health extensions workers as well as those who have left the program. By including HEW leavers, this study offers an additional perspective on some of the reasons that CHWs may leave their position. - The framework that was adapted from Ormel et al was a useful way to organize the types of factors that affect CHW motivation. - The study adds to existing evidence that both financial and non-financial factors are important for CHW motivation in their role. Comments: Introduction 1. I would have liked more description about the social identity approach that the authors are using (Lines 409-416) earlier on in the paper- maybe in the introduction. It was mentioned in the discussion, but it wasn't clear how it was connected with the framework that was being used. 2. In the introduction, the authors note that HEWs have poor retention (Line 117). If available, it would be helpful to include information on the percentage of HEWs in Ethiopia that leave every year. Methods 3. What was the motivation for only doing interviews and not also, for example, doing focus group discussions? 4. How were people recruited for this study? For the active HEW's, the authors say they used maximum variation sampling. Was this from a list of HEWs? How did they find the pool of HEWs from which they sampled?
--

	5. How did the authors decide on the sample size for each group of people that they interviewed? 6. The authors note that 2 interviews were not audio-recorded. Could the authors provide more detail on why this was the case and what implications it had for transcription and analysis of the data? Results 7. While the authors discuss the age and years of experience for the sample, it might be helpful to have a little more detail on the demographic characteristics of HEWs (both active and leavers), if this was collected. For example, what is their education level? What is their marital status and do most of them have children? Do they have other occupations as well? Also, it would be useful to also include the range for age and years of experience. 8. Given that both active HEWs and leavers were interviewed in this study, I would have liked some discussion on any differences in the motivations of current HEWs and those who had left. Could the results speak to why some HEWs have stayed while others have left? Or was it just that the leavers had experienced a particular event that had caused them to quit? 9. There was little discussion of the findings from the key informant interviews (apart from noting that they also felt that the financial incentives were insufficient). Did their perceptions of the motivations of HEWs generally concur with what the HEWs reported themselves? Did they have any other insights into why some HEWs quit their role? 10. In the results section the authors discuss some catalysts for HEWs quitting their role. Since these are separate from HEW motivations it might be helpful to see how they would be included in the framework in Figure 1. Discussion 11. It would be useful for the discussion section to include more policy implications of the study results. What kind of interventions do the authors propose implementing to reduce turnover of HEWs based on this study? For example, how can we ensure better supervision of HEWs or more positive community attitudes? And are these interventions necessarily simpler and more cost-effective than providing larger financial incentives? 12. The authors finding that recognition from the community and supervisors is an important motivator for CHWs has also been found in other studies on CHW motivation and it would be useful to contextualize their findings within this literature.
--	--

VERSION 1 – AUTHOR RESPONSE

Reviewer 1: Ruth Jackson

Comment 1: In the title, should it read ...A qualitative case study in Ethiopia rather than of Ethiopia?

Response: Thank you for this comment. We have changed the manuscript title to
“Understanding the importance of non-material factors in retaining community health workers in low-income settings: A qualitative case-study in Ethiopia”

Comment 2: In the Abstract, spell out Health Extension Workers the first time you use

Response: Thanks, this change has now been made, please refer to **line 47**.

Comment 3: Although you provide the total number of HEWs, can you provide evidence of how many new HEWs are being trained each year and what the attrition rate actually is? How many are being offered upgrading training to Level IV each year? Attrition is a significant problem, but you don't quantify it.

Response: Thanks, we have now included recent findings from an unpublished report on the national assessment of HEP. Lines 116 -124 read

“A recent, national evaluation of HEP had found the overall job satisfaction of HEWs to be quite low. More than half of the study sample reported to be unsatisfied with their current posts, suggesting that their retention could be affected in the long run. These apprehensions were substantiated by data indicating a gradual rise in the rate of attrition amongst HEWs over the programmatic lifetime of HEP, between 2005 and 2019. The average annual rate of attrition was reported to be close to 3%, with overall attrition since the start of the program being 21%. (22) This showed a clear rise in HEW attrition since the last national assessment of HEP published in 2011, which estimated overall attrition in the cadre for the period between 2005 and 2010 to be 6.5% (23)”

Unfortunately, to the best of our knowledge, there are no studies that report the numbers of HEW trainees leaving coaching institutions each year.

Comment 4: p. 7. Where in Tigray and SNNPR was the study conducted? Both are very large regions with considerable geographical differences. Even if you don't want to name the woredas, perhaps indicate the zones or give some idea where the study was located in the regions. Were HEWs from rural, remote or urban kebeles/woredas?

Response: Thank you for noting this. We conducted the study in four districts, two each in Tigray and SNNPR and had earlier refrained from mentioning their names as no substantial difference was noted between districts. The names have now been included, in addition to the fact that respondents were from a mix of rural and urban areas. Please see lines 163-169. **“Qualitative in-depth interview data were collected from two districts, each in two Ethiopian regions: Tigray and Southern Nations Nationalities and People's Republic (SNNPR), between May and July 2019. Raya Azebo, Saharti Samra districts were included in Tigray and Dilla Zuria, Silte in SNNPR. The respondents were located in a mix of urban and rural settings. The regions were purposively sampled to capture the varying extent of HEP implementation and thus HEW retention in different regions, which was likely to differ due to the political set up in Ethiopia.”**

Comment 5: There's a lot of research on the role and performance of HEWs, but far less research on the experiences of HEWs themselves. Are you able to provide a bit more detail and description about the HEWs? How far was the health centre from the health post where they worked? Were they supported by the Health Development Army in their area or did they find it added to their workload? Has the HDA strengthened HEWs relationship with the community? What about expectations about being involved in politics? Being on the kebele council? Just a couple of extra sentences perhaps so reader get more of a feel about them.

Response: Thank you. We refrain from going too deeply into the general experiences of HEWs because our main focus is understanding material and non-material incentives

affecting CHW retention and because of a limited word count have had to select which themes to elaborate. For example, in lines 296-299 we mention that their physical burden was an issue due to the lack of transportation to work areas but since their travel and distance between the health post and health centre did not emerge as a dominant theme, we don't elaborate on it. Regarding HDAs, our data does not show that HEW relationships with HDAs influenced their preferences about jobs. In addition, since HDAs are unique to the Ethiopian health system and our purpose was to add to the global debate on motivation and retention of CHWs, we decided not to go in much detail there. Regarding politics, we did have informal conversations with HEWs who mentioned their displeasure about having to join local political organisations in some cases, but this was not a prominent theme reported in our transcribed interviews, and thus not included.

Comment 6: On the role of gender, female HEWs are subject to the same social hierarchies that prevail in the wider society so relying on female HEWs may replicate or reinforce gender norms and biases in communities and within health systems. (Maes K, Closser S, Vorel E, Tesfaye Y. Using community health workers. Discipline and Hierarchy in Ethiopia's Women's Development Army Ann. Anthropol Pract. 2015;39(1):42–57) and see note 8. Consider whether you should mention the top-down nature of Ethiopia's HEP, and the fact that health workers such as HEWs are expected to remain 'subordinate'. Were HEWs free to move around?

Response: Thanks, while we recognise that gender hierarchies within HEP can be complicated, we do not have sufficient data from our interviews to suggest that HEWs are expected to remain subordinate, and hence would refrain from including this in the paper. We have however elaborated our discussion on need for gender transformation in their work environment. Lines 445-452 read **"In a Cochrane review, socio-cultural norms that restrict movement of female CHWs and govern acceptable male-female communications were also identified as barriers to doing their jobs successfully (47). Jackson et al (48) apply a gender lens to HEP and to the role of HEWs, and conclude that by changing gender norms and reducing constraints to gender equality, HEP could have more transformative outcomes not just for HEWs but for the communities they serve. Since majority HEWs in Ethiopia are women of reproductive age, providing them with childcare, particularly for when they are away for house visits, could be a step forward in gender transforming their work environment."**

Comment 7: You could add that you tried to avoid/reduce social desirability bias (in addition to) conducting interviews in private and by ensuring HEWs understood that interviews were confidential. It is well known that responses can often be 'in line with government policy' or that respondents may not want to raise negative or politically sensitive issues.

Response: That's a great point. Our training for research assistants who carried out these interviews with respondents did include key points on how to identify and address social desirability tendencies in respondents. The training material covered many of the points raised in the following paper: Bergen N, Labonté R. "Everything Is Perfect, and We Have No Problems": Detecting and Limiting Social Desirability Bias in Qualitative Research. Qualitative Health Research. 2020 Apr;30(5):783-92.

Lines 197-199 now mention the inclusion of this during enumerator training and reads **"Training included how to identify and reduce social desirability tendencies in respondents."**

Comment 8: A paper you may have missed is this one:
Jackson, R., Kilsby, D., & Hailemariam, A. (2019). Gender exploitative and gender transformative

aspects of employing Health Extension Workers under Ethiopia's Health Extension Program. *Tropical Medicine & International Health*, 24(3), pp. 304-319. doi:10.1111/tmi.13197

Although the focus is on gender, much of the data are similar. The paper also helps provide a better understanding of the working conditions of HEWs, lack of opportunities for training, safety, recognition, trust and respect from community members, and reasons why HEWs want to leave their positions.

Response: Thanks for the reference, it's a very informative paper. We have included the following text in line with this paper in Lines 447-450 "**Jackson et al (48) apply a gender lens to HEP and to the role of HEWs and conclude that by changing gender norms and reducing constraints to gender equality, HEP could have more transformative outcomes not just for HEWs but for the communities they serve**"

Comment 9: Study limitations. Can the outcomes of the study be generalised to other regions in Ethiopia? Or other sub-Saharan countries? If not, you can mention that generalisability wasn't the intention but that your study is unique as it captures the voices and experiences of HEWs

Response: Since community health worker cadres are so differently positioned in different country contexts, our findings may not be generalisable as is for other sub-Saharan countries. However, we do think that lessons can be learnt, and our study contributes new empirical evidence to the global debate on factors influencing the motivation and retention of CHWs. This is included in lines 401-403 and reads "**Our study findings from two regions in Ethiopia contribute new empirical evidence to the global knowledge and debate on factors influencing the motivation and retention of CHWs in LMICs.**"

Regarding other regions in Ethiopia, we believe that our findings may be generalised to other regions of Ethiopia, particularly those that are contextually similar to our study areas.

Comment 10: Page 3, 'Interviews were audio-recorded, and simultaneously translated into English and transcribed for analysis.' This sentence needs tweaking: Interviews were audio-recorded and then simultaneously ...

Response: Thanks for this correction. Changes were made to **line 35**.

It now reads '**Interviews were audio-recorded, and then simultaneously translated into English and transcribed for analysis.**'

Comment 11: Page 9. Line 182. Just to be clear, They were all women...I'm sure you mean the research assistants but there's a lot of going back and forth between research assistants and respondents in this paragraph and I had to read it again to be sure so consider if you need to clarify.

You don't mention how you recorded information for the two interviews that were not audio recorded. Did you take notes?

In my experience, health posts are not necessarily private spaces as people are often coming and going to see the HEWs and there are usually two HEWs assigned to each health post. How did you ensure the health post was a private space for you to interview one HEW? How did you select one HEWs at a health post (and not the other, or did you interview two HEWs at the same health post?).

Response: Thanks for this correction, changes were made to lines 198,199 It now reads "**All research assistants were Ethiopian women between ages 24 and 35 years**". We have also now made explicit how we recorded and analyzed the two interviews that could not be audio-recorded. Lines 209-212 have been changed to "**For the two interviews where audio**

recording was not possible due to respondent refusal, research assistants took down detailed notes which were later used for analysis”

Since HEWs were purposively sampled for diversity of age, experience etc it was not always required to interview both HEWs. In addition, both HEWs were often not available at the health post at the same time because as per protocol one was normally out in the field. For each interview we were able to find space either in or around the health post - often at the back side of the post, where the respondent and the interviewer could be left alone. Lines 203-207 in the method section have been amended to read **“Each interview was conducted in the language local to that region, in private spaces - normally at the back side of the health post where the respondent and interviewer could be left alone. As much as possible, the research assistants held interviews when HEWs were comparatively less busy with work, and took on average 45 minutes to complete”**

Comment 12: Can you mention the attrition rate of other health professionals? I wonder if you can compare the attrition rates compared to other health staff. Perhaps doctors have more opportunities to go overseas? The 2010 Human Resources for Health Country Profile: Ethiopia, though out of date showed that that higher levels of the health workforce (such as general practitioners and specialists) are mainly dominated by males and nurses, midwives, and HEWs are predominantly female so is it gendered?

I have heard about HEWs going to work in the Middle East before, but given the clampdown on this by the government, I wonder just how many opportunities they might have to work elsewhere, unless they move to an urban centre.

There are a couple more references you may have missed and might like to consider here or in future publications for comparison.

This one looks at deployment and attrition of health workers in East Wollega:

<https://www.ncbi.nlm.nih.gov/pmc/articles/PMC3275894/>

Although this paper is from 2007, but you could make a comparison to your results as it mentions future aspirations of HEWs:

https://www.researchgate.net/publication/242466722_Study_of_the_Working_Conditions_of_Health_Extension_Workers_in_Ethiopia

Very few expect to stay in the kebele of their present assignment or even as a health extension worker for more than two years. Only 16% expect to stay more than three years. The majority would like/expect to upgrade to nurse (about 70%) and the rest to environmental health. Few mention pharmacy technician, administrative positions and, interestingly, two mentioned upgrading to diploma in HEW. None chose to move to private/NGO health services, working outside the health sector or stopping employed work altogether.

You might want to consider these reports as well (although it appears the links are no longer working you might be able to find them):

Center for National Health Development in Ethiopia, & Columbia University. (2011). Ethiopia Health Extension Program Evaluation Study, 2007-2010, Volume-II. Health post and HEWs performance Survey. Addis Ababa: <http://www.cnhde.org.et/wp-content/uploads/2013/04/Part-II.pdf>

From the report:

HEWs were asked to identify measures that could motivate HEWs and improve HEP services. The top five suggestions were: access to and provision of upgrading/promotion (57.5%), adequate supply of logistics (29.8%), salary increment (29.5%), access to transportation (15.4%), and regular supportive supervision (13%). Other factors suggested include refresher training and provision of

housing.

And this one:

Center for National Health Development in Ethiopia, & Columbia University. (2011).

Ethiopia Health Extension Program Evaluation Study, 2005-2010, Volume-IV. Support and management of HEP. Addis Ababa, Ethiopia: <http://www.cnhde.org.et/wp-content/uploads/2013/04/Part-IV.pdf>

Out of the total 3,241 HEWs deployed in the 64 woredas since HEP implementation, which varies between one to 6 years of implementation in the sample woredas, a total of 212 HEWs left their HEP work with overall attrition rate of 6.5%. The main reasons for leaving their HEP work in the woreda were: changed field of work (71 HEWs), due to personal reasons such as marriage and illness (68 HEWs), and due to uncomfortable work environment such as remoteness of kebeles, high workload, and low remuneration (31 HEWs). The total number of HEWs deployed since HEP was launched (one to five years of implementation period) in the 64 woredas surveyed was 3,241. The average number of HEWs deployed was 51 per woreda. Over the period of HEP implementation in the sample woredas, a total of 212 HEWs left their HEP work in the woreda with an average of 3 HEWs per woreda. The overall attrition rate was 6.5% over the program period. It should be noted that the average duration of implementation in Tigray and Dire Dawa is 5 years, and it is 4 years in Amhara and SNNP. In Oromia it is 2.7 years. Thus, the overall attrition rate of HEWs per year is about 2% (6.5% over 3.5 years).

Among the 212 HEWs who were reported to have left from assigned kebeles in their respective woreda, nearly a third (71 HEWs) left their kebele because they changed their field of work. An equivalent number (68) of HEWs left their job for personal reasons including marriage and medical reasons. The reason for some HEWs (31 HEWs) was reported to be uncomfortable work environment such as remoteness of kebele, workload, and low remuneration. The other reasons were because they were transferred to another woreda and dismissed due to discipline reasons.

Response: Thank you, we really appreciate the reviewer's detailed and insightful comments. We have now included data on attrition from a recent national evaluation of the health extension program, as mentioned in the comment above.

"Thanks for inputting about missed references as well. We have included the following text in lines 421-231" **For example, in Ethiopia, there is evidence that majority of HEWs are keen to take on more responsibility and upgrade to become nurses, pharmacy technicians and health administrators (40) but no such career path is offered to them. Access to and provision of upgrading and promotion opportunities was identified as one of top five measures to motivate HEWs and improve HEP services in another study (41). The need for and the value of career progression amongst CHWs to improve job satisfaction, was also a key topic of discussion at an international symposium on CHWs in 2019 (42). Moreover, the rate of attrition was reported to be nearly three times greater for higher level professionals like doctors and nurses, in comparison to other allied health professionals like HEWs, lower in the hierarchy. Factors identified as contributing towards these high attrition rates were budget constraints, but also the lack of educational opportunities and poor career development."**

On the reviewer's point about the CHW cadre in Ethiopia being gendered; we agree that it is especially as the health extension program envisaged the entire cadre to be female because their main tasks focussed on mothers and new-borns. However, this again was not a key theme and so we would refrain from going any deeper into this debate than we already have.

Comment 13: Recommendations. Are you able to make recommendations to policy makers 'that appeal to the social needs of CHWs could represent a simpler and more cost-effective means of improving their retention' (as you've stated in the Abstract) or about any of these any of these factors:

as Maes et al. 2019 have done: fair pay levels, opportunities for advancement, and representation in high-level policy decisions for would both motivate and truly empower them; or equal access to strategic resources such as mentoring and supervision, administrative and infrastructural support, secure funding sources and employment contracts and networking, as Jackson et. al 2019 have done?

Should HEWs be from the community that they serve? What about task shifting to the Health Development Army and the impact of this on the workload of HEWs? Has that helped increase acceptance and validation from the community? How can HEWs be better supported by their supervisors (who are often male)?

When getting to know, working with and interviewing HEWs over many weeks (and there were some HEWs I'd met years earlier doing other research), I sensed that once they had come to a decision to leave, they were going to leave no matter what, even if it meant for example, fulfilling the obligation to work for another year or two to fulfil their upgrading conditions. Awhile back I ran into a HEW who had told me she was 'definitely' going to leave, but she had finally managed (after years of requesting) to get a transfer to another kebele after she completed Level IV training. Another HEW had left to set up a shop in town with her family (she was from the original batch of HEWs from 2004, had worked for well over 10 years and was a bit older than the others), and another left to live with her husband in another woreda as the choice between being a HEW and being able to live with her husband and family became too difficult and she couldn't get a transfer.

Response: We really appreciate the reviewer's efforts and insightful comments in helping us improve this manuscript. We have considered their suggestions, and in line with them have added the following recommendations in lines 481-494" **It has already been recognised that innovative ways to support CHWs can contribute substantially in creating a more satisfied health workforce that is able and willing to continue delivering quality healthcare to communities (30,48, 55, 56). In the Ethiopian context, focus particularly could be accorded to improve not just the availability of strategic resources such as mentoring and supervision, but the quality of support offered by often male supervisors to these female workers. Addressing HEW aspirations to progress in their jobs by providing sufficient upgrading opportunities, tailored to their preferences and abilities, has good potential of improving their job satisfaction, reducing attrition. Clearly, positive community attitude towards HEWs is a key demand-side requirement for HEWs to stay motivated. We believe a good rapport between HEWs and the community often results when HEWs are capable of providing healthcare to the standards expected by the community. This of course is a function of having health posts with adequate infrastructure, well trained HEWs, but equally importantly emotionally satisfied HEWs who have not necessarily had to separate from their families due to lack of transfer opportunities."**

Comment 14: Future research. Can you make some suggestions for future research?

A couple of suggestions could come from Maryse Kok's publication, Performance of Community Health Workers: Optimizing the benefits of their unique position between communities and the health sector. She states that the study 'did not find many mechanisms related to feelings or behaviours of actors in the health sector that led to trusting relationships between them and CHWs. This could be an interesting topic for further research, to input into the development of interventions that stimulate trust between CHWs and actors in the health sector.'

She goes on to say it would be interesting to 'evaluate whether programmes that are designed with input from CHWs and community representatives are better tailored to the realities and needs of communities than programmes that are designed in a more top--down way.'

She also suggests that attention to 'outcome measure, such as competencies or self-esteem (presumably much more complex) is needed, preferably combined with outcomes at the level of the end-users or impact.'

Response: Thank you, we have included the following text on future research in lines 496-502 “ **Future research should explore the development of interventions that can create and maintain trust between CHWs and the community. It could further be evaluated if a bottom-up approach that is designed with the inputs of CHWs and the community is better tailored to the needs and realities of them both (27). In addition to health outcomes, policymakers should also invest in studying outcome measures such as competencies and self-esteem of health workers as that can have direct effects to their retention and indirect effects on the sustainable delivery of population health. “**

Reviewer 2: Sonia Hines

Comment 1: Line 117 The link between work satisfaction and retention has not been well made. The authors, if at all possible, should use actual figures of vacant positions, proportions of positions vacated over time, or other standard retention measures to illustrate the problem and justify the conduct of this study.

Response: We thank the reviewer for their comments. We have included results from a recent evaluation of the health extension program to make the link between retention and job satisfaction clearer. Lines 116-124 read “ **A recent, national evaluation of HEP had found the overall job satisfaction of HEWs to be quite low. More than half of the study sample reported to be unsatisfied with their current posts, suggesting that their retention could be affected in the long run. These apprehensions were substantiated by data indicating a gradual rise in the rate of attrition amongst HEWs over the programmatic lifetime of HEP, between 2005 and 2019. The average annual rate of attrition was reported to be close to 3%, with overall attrition since the start of the program being 21%. (22) This showed a clear rise in HEW attrition since the national assessment of HEP published in 2011, which estimated overall attrition in the cadre for the period between 2005 and 2010 to be 6.5% (23)”**

Comment 2: Lines 200-365 Overall the results section is well written and clear, but I think it would benefit with greater focus on participants' voices as illustrations of the findings the authors are making. Additionally, it would be useful to have a slightly more detailed description of each speaker being quoted, so the reader can get a sense of the variety of perspectives being shared.

Response: We thank the reviewer for their comments. We have made a conscious decision to have one or two quotes per theme, to allow enough space in the main document to be able to provide an interpretation for the reader. In addition to mentioning the group to which the respondents belonged and their residential region, we had also included their age in every quote. The journal needed explicit consent from individual respondents to include potentially identifiable information such as their age, so we chose to remove this information.

Comment 3: Several of the factors being reported are not being explicitly linked to retention by the authors. Financial incentives, material factors, and non-material factors in particular are presented like factors influencing job satisfaction and do not link to why the participant left or is considering leaving. A revision of the results section to refocus on the purpose of the study would be helpful.

Response: Thanks for the comment. We believe job satisfaction and motivation is closely related to the retention of these health workers and is demonstrated by the framework in

figure 1. The framework make it clear that factors like financial incentives, material and non-material factors do indirectly affect retention of HEWs.

Comment 4: Line 377 Referring to this research as 'case study' is inaccurate.

Response: Thanks, we have changed this in line 417 to **“our study offers a number of ...”**

Comment 5: Line 425 The point about the INSCALE project should be expanded to explain how it relates to this study

Response: Thanks, in line with your comment we have modified lines 478-491 to read **“Our study drew on formative research results from the inSCALE study and applied the social identity approach for establishing links between identification and motivation (52, 53) in the context of CHWs in Ethiopia.”**

Reviewer 3: Indrani Saran

Comment 1: INTRO: I would have liked more description about the social identity approach that the authors are using (Lines 409-416) earlier on in the paper- maybe in the introduction. It was mentioned in the discussion, but it wasn't clear how it was connected with the framework that was being used.

Response: Thanks, we have now added a description of the social identity approach in the introduction section. Lines 135-141 have been modified to read **“While material incentives that align with the preferences of CHWs are relevant to studying retention, behavioural theories like the social identity approach have seldom been applied to empirical findings, to account for the social behaviour of health workers. This approach closely studies the social identity, the context in which they work, and how certain factors could influence their labour choices to leave or stay in their jobs”**

Comment 2: In the introduction, the authors note that HEWs have poor retention (Line 117). If available, it would be helpful to include information on the percentage of HEWs in Ethiopia that leave every year.

Response: Thanks, we have now included recent findings from an unpublished report on the national assessment of HEP. Lines 116-124 read

“A recent, national evaluation of HEP had found the overall job satisfaction of HEWs to be quite low. More than half of the study sample reported to be unsatisfied with their current posts, suggesting that their retention could be affected in the long run. These apprehensions were substantiated by data indicating a gradual rise in the rate of attrition amongst HEWs over the programmatic lifetime of HEP, between 2005 and 2019. The average annual rate of attrition was reported to be close to 3%, with overall attrition since the start of the program being 21%. (22) This showed a clear rise in HEW attrition since the national assessment of HEP published in 2011, which estimated overall attrition in the cadre for the period between 2005 and 2010 to be 6.5% (23)”

Comment 3: METHODS: What was the motivation for only doing interviews and not also, for example, doing focus group discussions?

Response: We refrained from doing focus group discussions with study respondents because we wanted to capture individual experiences, perceptions and behaviours which could only be attained through in-depth interviews. Moreover, we believed that reasons why

people leave their jobs or things they find unsatisfying are of sensitive nature and unlikely to be disclosed in a group of peers.

This rationale is now included in lines 403-408 which reads **“Since we wanted to capture the individual experiences and behaviour of all three respondent groups, we refrained from conducting focus group discussions and stuck to using in-depth interviews. Moreover, we believed that reasons why people leave their jobs or things they find unsatisfying are of sensitive nature and unlikely to be disclosed in a group of peers”**

Comment 4: How were people recruited for this study? For the active HEW's, the authors say they used maximum variation sampling. Was this from a list of HEWs? How did they find the pool of HEWs from which they sampled?

Response: Yes, active HEWs were selected from a list with names of those working in the selected study districts. This information is now included in table 1.

Comment 5: Methods: How did the authors decide on the sample size for each group of people that they interviewed?

Response: We interviewed respondents till sufficient saturation in themes emerging from interview data was attained. This information has now been added to lines 179, 180. It now reads **“We interviewed respondents till sufficient saturation was attained in the themes emerging from interview data. A total of 47 semi-structured interviews were conducted”**

Comment 6: The authors note that 2 interviews were not audio-recorded. Could the authors provide more detail on why this was the case and what implications it had for transcription and analysis of the data?

Response: Audio recording was not possible due to respondent refusal. For these two, research assistants took down detailed notes which were later used as transcripts during data analysis. This information has now been added to lines 209-211 It reads **“For the two interviews where audio recording was not possible due to respondent refusal, research assistants took down detailed notes which were later used for analysis”**

Comment 7: Results - While the authors discuss the age and years of experience for the sample, it might be helpful to have a little more detail on the demographic characteristics of HEWs (both active and leavers), if this was collected. For example, what is their education level? What is their marital status and do most of them have children? Do they have other occupations as well? Also, it would be useful to also include the range for age and years of experience.

Response: Thanks, after evaluating the completeness of social demographic characteristics that were collected, we were able to add the marital age of participants in row 4 of table 1. The spread for age and years of experience are now included in lines 180-185. **“The mean age of respondents across all three study groups was 31 years, ranging from 24 to 40 for leavers, 20 to 49 for HEWs and 26 to 48 for KIs. The mean time worked in the health system was 6 years for HEWs, spread between 1 and 13 years and 7 years for key informants, ranging from 0.5 to 8 years. Leavers had spent on average 6 years in their current jobs, ranging from 1 to 9 years”**

Additional demographic details such as a detailed description of the occupations of leavers is given in the results section under “leavers destinations”.

Comment 8: Results - Given that both active HEWs and leavers were interviewed in this study, I would have liked some discussion on any differences in the motivations of current HEWs and those

who had left. Could the results speak to why some HEWs have stayed while others have left? Or was it just that the leavers had experienced a particular event that had caused them to quit?

Response: Thanks for this. We describe in detail why some HEWs left their jobs while others stayed in the section titled “Catalysts influencing exit: pro-social preferences”, from line 290. We found that while HEWs did not generally anticipate leaving their jobs, they did leave when they felt they had lost the appreciation of their community or supervisors. We call these triggers ‘catalysts’ that speeded up their attrition from positions.

Comment 9: Results - there was little discussion of the findings from the key informant interviews (apart from noting that they also felt that the financial incentives were insufficient). Did their perceptions of the motivations of HEWs generally concur with what the HEWs reported themselves? Did they have any other insights into why some HEWs quit their role?

Response: Thanks for this. We felt that the data from KIs did not add additional themes on why HEWs leave and so we decided to focus on quoting the views and experiences of HEWs and leavers themselves, rather than the KIs. The dominant theme on reasons for attrition from KIs was in fact the issue of financial incentives being insufficient, which we have included in results.

Comment 10: In the results section the authors discuss some catalysts for HEWs quitting their role. Since these are separate from HEW motivations it might be helpful to see how they would be included in the framework in Figure 1.

Response: Thanks. In the original manuscript the section on catalysts called “catalysts influencing exit: pro-social preferences” is used to indicate that they fit best under *extrinsic motivational factors*, particularly *non-material incentives: pro-social preferences*, included in figure 1.

Comment 11: Discussion: It would be useful for the discussion section to include more policy implications of the study results. What kind of interventions do the authors propose implementing to reduce turnover of HEWs based on this study? For example, how can we ensure better supervision of HEWs or more positive community attitudes? And are these interventions necessarily simpler and more cost-effective than providing larger financial incentives?

Response: Thank you, the points the reviewer has raised are worth shining light on. We have now included recommendations in lines 481-502. It reads **“It has already been recognised that non-material interventions to support CHWs can contribute substantially in creating a more satisfied health workforce that is able and willing to continue delivering quality healthcare to communities (30,48,55,56). In the Ethiopian context, focus particularly could be accorded to improve not just the availability of strategic resources such as mentoring and supervision, but the quality of support offered by often male supervisors to these female workers. Addressing HEW aspirations to progress in their jobs by providing sufficient upgrading opportunities, tailored to their preferences and abilities, has good potential of improving their job satisfaction, reducing attrition. Clearly, positive community attitude towards HEWs is a key demand-side requirement for HEWs to stay motivated. We believe a good rapport between HEWs and the community often results when HEWs are capable of providing healthcare to the standards expected by the community. This of course is a function of having health posts with adequate infrastructure, well trained HEWs, but equally importantly emotionally satisfied HEWs who have not necessarily had to separate from their families due to lack of transfer opportunities.**

Future research should explore the development of interventions that can create and maintain trust between CHWs and the community. It could further be evaluated if a bottom-up approach that is designed with the inputs of CHWs and the community, is better tailored to the needs and realities of them both (27). In addition to health outcomes, policymakers should also invest in studying outcome measures such as competencies and self-esteem of health workers as that can have direct effects to their retention and indirect effects on the sustainable delivery of population health.”

Comment 12: The authors finding that recognition from the community and supervisors is an important motivator for CHWs has also been found in other studies on CHW motivation and it would be useful to contextualize their findings within this literature

Response: Thank you, we have acknowledged this literature in lines 481-484 “**It has already been recognised that non-material interventions to support CHWs can contribute substantially in creating a more satisfied health workforce that is able and willing to continue delivering quality healthcare to communities. (30,48,55,56).**”

VERSION 2 – REVIEW

REVIEWER	Ruth Jackson School of Humanities and Social Sciences Faculty of Arts and Education Deakin University Australia
REVIEW RETURNED	22-Aug-2020

GENERAL COMMENTS	The authors have addressed all the reviewer's comments. In the copy edit stage there are a couple of minor things to fix. A couple of obvious ones are: Reference No 20 Transformation is misspelt Page 5 Line 94 remove the comma so it reads, 'While there is some limited...' Page 21, Lines 492 to 494 the sentence is possibly a bit verbose so maybe it can be rewritten I'm unsure why a couple of changes were made in the Abstract. Can you rethink these: The motivation and retention...(you've replaced 'and' with a comma but I think it should still be and) Should it be?: The motivation and retention...is a challenging (not challenge) You've also removed the word 'and' and replaced with a comma between extrinsic and intrinsic to extrinsic, intrinsic...perhaps reconsider whether to use 'and'. This is a high quality paper that has implications that I hope will be considered by policymakers seeking ways to improve the working conditions of HEWs and CHWs. As the authors conclude, it is crucial to listen to the voices of health workers while designing programs to improve retention.
--

REVIEWER	Sonia Hines Centre for Remote Health, Flinders University, Australia
REVIEW RETURNED	12-Aug-2020

GENERAL COMMENTS	Thank you for your revisions, they have significantly improved the reporting of this interesting study.
---

REVIEWER	Indrani Saran Boston College, United States of America
-----------------	---

REVIEW RETURNED	24-Aug-2020
-------------

GENERAL COMMENTS	Thank you for the thorough response to the comments. I have a few remaining minor comments:  1. Line 39: I think the comma is mis-placed as it is not clear if there are 2 districts in 2 regions (total of 2 districts), or 2 districts each in 2 regions (total of 4 districts). I believe it is the latter, but the way it is written makes it sound like the former. 2. Line 43 (abstract): HEW should be written out the first time the acronym is used (it is spelled out later). 3. In lines 57-58, and again in lines 510-513 the authors suggest that appealing to the social needs of CHWs could be “cost-effective and simpler”. It’s not clear that this is necessarily the case since ensuring good quality supervision, providing infrastructure and supplies to improve relations with the community and creating good career progression could potentially be both costly and difficult to implement. 4. Lines 96 and Lines 131: There seem to be mis-placed commas in these sentences. 5. Line 138: the reference to “certain factors” seems a little vague. I liked the description of the social identity approach provided in the Discussion (lines 462-467), and think it would be helpful if that was brought to the Introduction. 6. In Lines 169-171 the authors note that Tigray has better health indicators, does it also have higher HEW retention? 7. In Lines 184-185, the authors provide information on how long leavers had spent in their current job, do they also have information on how long leavers spent as HEWs? 8. Lines 427-429: It was not clear to me how the attrition rate for doctors and nurses was relevant for HEWs and their career progression.
---

VERSION 2 – AUTHOR RESPONSE

Reviewer 2: Ruth Jackson

Comment 1: Reference No 20: Transformation is misspelt

Response: Thanks, this has now been corrected on line 606

Comment 2: Page 5 Line 94 remove the comma so it reads, 'While there is some limited...'

Response: Thanks, we have made this change on line 94

Comment 3: Page 21, Lines 492 to 494 the sentence is possibly a bit verbose so maybe it can be rewritten .

Response: Thanks, we have re-written the sentence and lines 492-495 now read

“This of course is a function of having health posts with adequate infrastructure as well as well-trained HEWs. Equally important is that HEWs are emotionally satisfied, not having to live apart from their families due to the lack of transfer opportunities”

Reviewer 3: Indrani Saran

Comment 1: Line 39: I think the comma is mis-placed as it is not clear if there are 2 districts in 2 regions (total of 2 districts), or 2 districts each in 2 regions (total of 4 districts). I believe it is the latter, but the way it is written makes it sound like the former. .

Response: Thanks, the comma has now been removed in line 39.

Comment 2: Line 43 (abstract): HEW should be written out the first time the acronym is used (it is spelled out later).

Response: Thanks, these changes have now been made on lines 43 and 47.

Comment 3: In lines 57-58, and again in lines 510-513 the authors suggest that appealing to the social needs of CHWs could be “cost-effective and simpler”. It’s not clear that this is necessarily the case since ensuring good quality supervision, providing infrastructure and supplies to improve relations with the community and creating good career progression could potentially be both costly and difficult to implement.

Response: Thank you. While we agree that these strategies are not always simpler to implement and necessarily cost-effective, we do believe that they are more acceptable and cost-effective in the long run as they create more resilient health systems. In line with your comment, we have now modified lines 56-58 to read

“Our findings suggest that policy interventions that appeal to the social needs of CHWs can prove to be more acceptable and potentially cost-effective in improving their retention in the long run.”

And lines 510-513 to read

“Thus, appealing to their social needs may represent a relatively more acceptable, potentially cost-effective and complementary strategy to the traditional approach of using financial incentive packages for improving retention, particularly in the long run in resource-constrained settings.”

Comment 4: Lines 96 and Lines 131: There seem to be mis-placed commas in these sentences

Response: Thanks, these commas have now been removed.

Comment 5: Line 138: the reference to “certain factors” seems a little vague. I liked the description of the social identity approach provided in the Discussion (lines 462-467), and think it would be helpful if that was brought to the Introduction

Response: Many thanks for your comment. We did not want to bring that paragraph in the discussion section forward to introduction as the social identity approach is not being dealt with in detail in introduction and just a brief introduction of the approach would have broken continuity. We have however modified lines 137-140 read

“This approach studies the social identity, context in which they work, along with self-efficacy and outcome expectancies that could influence their labour choices to stay in or leave their jobs. Further description of this approach is provided in the discussion section.”

Comment 6: In Lines 169-171 the authors note that Tigray has better health indicators, does it also have higher HEW retention?

Response: Thanks, to the best of our knowledge there are no studies that provide the magnitude of HEW attrition stratified by region, so we are not able to say where Tigray stands. However, along with colleagues in Ethiopia and using data from the recent HEP evaluation, we are currently in the process of developing a manuscript that provides information about HEW attrition stratified per region so this information should be in the public domain soon.

Comment 7: In Lines 184-185, the authors provide information on how long leavers had spent in their current job, do they also have information on how long leavers spent as HEWs?

Response: Thank you. Unfortunately this information about their current jobs was not available for a significant proportion of leavers so was not included.

Comment 8: Lines 427-429: It was not clear to me how the attrition rate for doctors and nurses was relevant for HEWs and their career progression

Response: Thank you. The suggestion to include attrition figures of other cadres of health workers in Ethiopia was provided by another reviewer and was therefore included.